# Maize Ear Height and Ear–Plant Height Ratio Estimation with LiDAR Data and Vertical Leaf Area Profile

**Han Wang** [1,2]**, Wangfei Zhang** [2]**, Guijun Yang** [1,3]**, Lei Lei** [1,3]**, Shaoyu Han** [1]**, Weimeng Xu** [1]**, Riqiang Chen** [1]**, Chengjian Zhang** [1] **and Hao Yang** [1,*]

1   Key Laboratory of Quantitative Remote Sensing in Agriculture of Ministry of Agriculture and Rural Affairs, Information Technology Research Center, Beijing Academy of Agriculture and Forestry Sciences, Beijing 100097, China
2   College of Forestry, Southwest Forestry University, Kunming 650224, China
3   College of Geological Engineering and Geomatics, Chang'an University, Xi'an 710054, China
*   Correspondence: yangh@nercita.org.cn

**Abstract:** Ear height (EH) and ear–plant height ratio (ER) are important agronomic traits in maize that directly affect nutrient utilization efficiency and lodging resistance and ultimately relate to maize yield. However, challenges in executing large-scale EH and ER measurements severely limit maize breeding programs. In this paper, we propose a novel, simple method for field monitoring of EH and ER based on the relationship between ear position and vertical leaf area profile. The vertical leaf area profile was estimated from Terrestrial Laser Scanner (TLS) and Drone Laser Scanner (DLS) data by applying the voxel-based point cloud method. The method was validated using two years of data collected from 128 field plots. The main factors affecting the accuracy were investigated, including the LiDAR platform, voxel size, and point cloud density. The EH using TLS data yielded $R^2 = 0.59$ and RMSE = 16.90 cm for 2019, $R^2 = 0.39$ and RMSE = 18.40 cm for 2021. In contrast, the EH using DLS data had an $R^2 = 0.54$ and RMSE = 18.00 cm for 2019, $R^2 = 0.46$ and RMSE = 26.50 cm for 2021 when the planting density was 67,500 plants/ha and below. The ER estimated using 2019 TLS data has $R^2 = 0.45$ and RMSE = 0.06. In summary, this paper proposed a simple method for measuring maize EH and ER in the field, the results will also offer insights into the structure-related traits of maize cultivars, further aiding selection in molecular breeding.

**Keywords:** ear height; ear–plant height ratio; TLS LiDAR; DLS LiDAR

## 1. Introduction

Increasing cereal yields is one way to address global food security [1]. The rapid development of maize breeding programs is the main approach to increase the yield of maize (*Zea mays* L.) [2]. Maize height traits (i.e., plant height, ear height, and ear–plant height ratio) have played an important role in improving maize lodging resistance and increasing grain yields [3]. Ear height (EH) and ear–plant height ratio (ER) affect maize morphology and yield mainly by influencing the architecture and centroid of individual plants [4,5].

EH is highly affected by planting density and leaf area index (LAI) [6]. Optimal EH is critical for improving population density and maximizing the utilization of fertilizer, moisture, and incident photosynthetically active radiation [7,8]. In addition, ER is considered an important parameter for evaluating lodging resistance and yield [9–11]. The optimal range of ER should be taken into account when breeding or selecting lodging-resistant and high-yielding hybrids as well as developing crop management practices that minimize the risk of lodging [9].

The current measurements of EH and ER mainly rely on traditional manual ground surveys [12]. However, manual measurements are labor-intensive and time-consuming [13,14],

and they are not suitable for large-scale and reproducible surveys. Therefore, there is an urgent need in breeding projects for developing simple maize EH and ER estimation approaches.

In recent years, some studies have been able to identify maize ears through image recognition, but they were carried out in the laboratory and could only estimate the EH of individual maize plants. For example, Ye et al. [15] chose the commonly used Histogram of Oriented Gradients/Support Vector Machine (HOG/SVM) detection framework to identify the ear of maize. Subsequently, Yu et al. [16] modified the method in [15] and obtained better identification of ear location results. In addition, some studies have developed maize ear position recognition algorithms at the field scale. Brichet et al. [17] developed a pipeline based on an ear detection algorithm and placed it on a phenotyping platform to evaluate the location of 60 maize hybrid ears. Despite these improvements, these algorithms are still unable to detect EH quickly and effectively over large areas.

Both EH and plant height (PH) are phenotypic parameters that reveal information about the vertical structure of the crop. Methods for PH estimation using remote sensing technology are mature. Common PH estimation methods can be divided into three categories according to the sensor used: RGB [18–20], multispectral [21], and light detection and ranging (LiDAR) [22,23]. Some studies use a combination of different types of data to estimate PH. Han et al. [24] proposed a plot-scale PH extraction method based on the spatial structure of the maize canopy using a combination of RGB and multispectral data. Gao et al. [25] estimated the individual PH of maize based on LiDAR and RGB images using the difference between the maximum elevation value of the LiDAR point cloud and the average elevation value of the bare digital terrain model. However, due to canopy occlusion and leaf overlaps that can occur in mature maize canopies, existing crop PH detection models are unsuitable for detecting EH. Additionally, an approach for the large-scale field estimation of maize EH still needs to be devised.

LiDAR is an active sensing technology that is not affected by natural lighting conditions and can measure the 3D canopy structure of plants with high precision [26,27]. LiDAR technology has been widely used to estimate crop structural parameters such as leaf area index (LAI) [26,28] and leaf area density (LAD) [27,29]. The LAI inversion using LiDAR is similar to the leaf area density (LAD) inversion model; both are usually obtained by applying porosity and voxel-based models [30–33]. Nie et al. [34] proposed a new height threshold method based on airborne discrete-return LiDAR data to separate ground returns from canopy returns and better estimate the LAI of maize. Lei et al. [35] used a hierarchical method to compare the effects of different incident angles on the LAI estimation accuracy, and the optimal voxel size was considered to depend on the characteristics of the LiDAR instrument used (such as incidence angle, flight height, flight speed, laser beam's diameter, and the range and resolution of the laser scanner, among others) and the characteristics of the experimental area; they finally determined the optimal voxel as 0.04–0.055 m. Jin et al. [36,37] proposed combining deep learning algorithms with geometric principles to extract leaf area from ground-based LiDAR data accurately. There are certain developments and outcomes of LAD estimation using LiDAR [38], which provide the basis for the rapid large-scale estimation of EH and ER of maize in the field.

Maize plant leaves are divided into four groups according to their position of node, characteristics, and physiological functions: (1) basal leaf (root leaf group); (2) lower leaf (stalk leaf group); (3) middle leaf (cob leaf group); and (4) upper leaf (grain leaf group). The leaf area at each node of the main stalk of maize varies depending on the variety, but in all cultivars, the leaf area is greatest in the middle leaf group. Three leaves (ear position leaf and its upper and lower leaves) are located in the middle leaf group. They have the longest, widest, and largest leaf area and their distribution facilitates the accumulation of dry matter in the cob [39]. As one of the canopy vertical structure parameters, the leaf area density (LAD) in each horizontal layer is generally used to quantify the leaves in the canopy [40]. The relationship between vertical leaf area distribution and height follows a bell-shaped distribution function [41], making it possible to estimate EH from the distribution curve of maize LAD.

This study proposes a simple estimation model of maize EH and ER using LiDAR data based on their relationship with maximum LAD at the ear. The model estimates LAD based on the voxel method, which is used to voxelize the point cloud data and uses the point quadratic model to calculate the contact frequency of the laser beam in each layer. The model, which uses the point cloud statistical information of the maize ear to obtain the average EH, does not require accurate identification and positioning of the ear position and is less affected by canopy occlusion. This paper evaluates the accuracy of the model on different platforms by comparing Terrestrial Laser Scanner (TLS) and Drone Laser Scanner (DLS) platforms and determines at what planting densities of maize the model is applicable.

## 2. Materials and Methods

### 2.1. Study Area and Experimental Designs

The study area is located in the Xiaotangshan National Precision Agriculture Research Demonstration Base (40°10′60″N, 116°26′30″E), Changping District, Beijing. The average annual precipitation in this area is 508 mm and the annual average temperature is 13 °C. In 2019 and 2021, we obtained LiDAR data of the maize filling period for different platforms and different sensors. The experiments were all designed with a completely randomized design. In 2019, the experiment consisted of five cultivars (A1–A5), four planting densities (D2–D5), and three random replicates. In 2021, the experiment consisted of five cultivars (A1, A6–A9), four planting densities (D1–D4), two planting row orientations, and two random replicates. The specific experimental design is shown in Table 1. In 2021, the study area was impacted by large hail, which lodged some of the plots. Therefore, only 68 plots with a small amount of lodging were used.

**Table 1.** Introduction to the experimental plan.

| Year | Variety | Density (Plant/ha) | Plots | Plot Length and Width (m) | Row Spacing (m) | Platform |
|---|---|---|---|---|---|---|
| 2019 | ZhengDan958 (A1), XianYu335 (A2), JingNongke728 (A3), ChengDan30 (A4), JingPin6 (A5) | 45,000 (D2), 67,500 (D3), 90,000 (D4), 105,000 (D5) | 5 cultivars * 4 densities * 3 random replicates = 60 | 3.6 × 2.5 | 0.6 (6 rows per plot) | DLS TLS |
| 2021 | ZhengDan958 (A1), JingJiu16 (A6), TianCi19 (A7), JingNuo2008 (A8), NongKeNuo336 (A9) | 33,000 (D1), 45,000 (D2), 67,500 (D3), 90,000 (D4) | 5 cultivars * 4 densities * 2 planting rows orientations * 2 random replicates = 80 | 3.6 × 2.5 | 0.6 (6 rows per plot) | DLS TLS |

Note: * indicates multiplication sign

### 2.2. Data Acquisition

#### 2.2.1. TLS Data

In the experiment, the field TLS data of the maize filling period were collected on 12 September 2019 and 22 August 2021. Data collection adopted the multi-station mode. The scanning stations were set up using the five-point method, distributed between four adjacent plots in every two columns (Figure 1d,e). We acquired data using the LiDAR FARO FocusS 350 sensor (version 2019.0.1.1653, FARO Scanner Production GmbH Co., Lake Mary, FL, USA). The specific parameters are shown in Table 2.

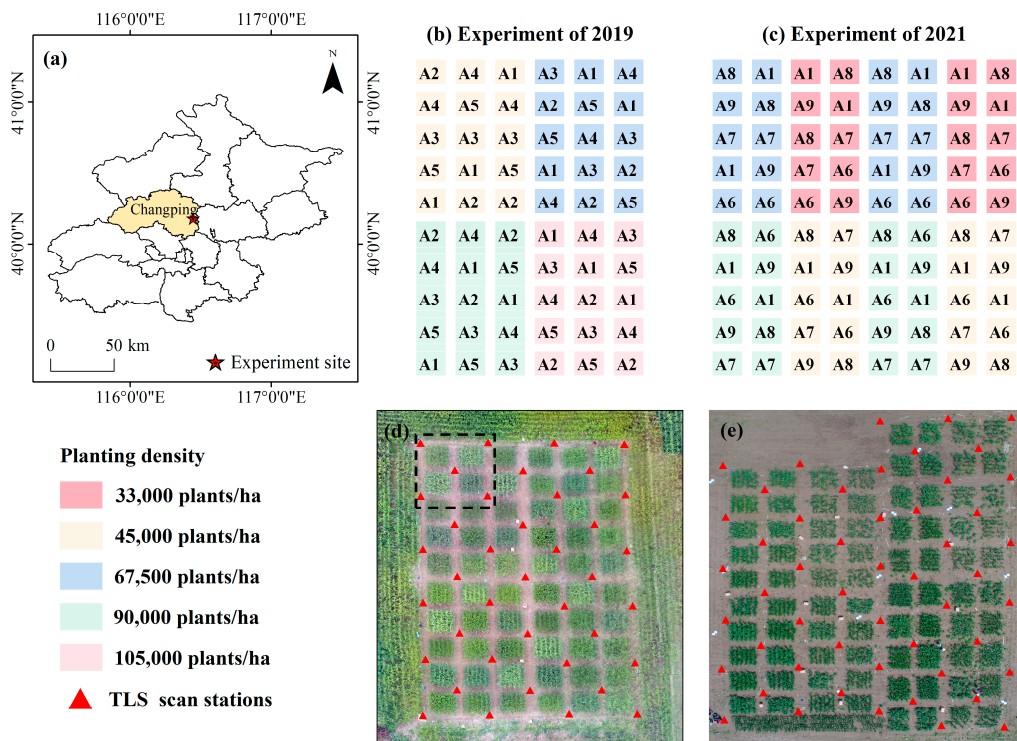

**Figure 1.** Study area and experimental design. (**a**) Study area; (**b**) map of variety density in the experimental area in 2019; (**c**) map of variety density in the experimental area in 2021; (**d**) orthophoto of maize experimental plot in 2019; (**e**) orthophoto of maize experimental plot in 2021.

**Table 2.** Main technical parameters of the LiDAR FARO FocusS 350 sensor.

| Parameters | Specification |
| --- | --- |
| Scanning distance | 20 m |
| Horizontal field angles | 0–360° |
| Vertical field angles | −60–90° |
| Scanning accuracy | ±1 mm |
| Scan time per station | less than 2 min and 54 s |
| Measurement speed | up to 976,000 points/s |
| Scanner height | 1.5 m |

### 2.2.2. DLS Data

The DLS data of the maize filling period were acquired on 12 September 2019 and 21 August 2021. EH had largely stabilized at the filling period. The DLS system used in the experiment included the following four main components: unmanned aerial vehicle (UAV) platform (DJI M600, DJI, Shenzhen, China), sensor (Riegl VUX-1, RIEGL Co., Lower Austria, Austria, with main parameters listed in Table 3), antenna (to acquire GPS satellite signals in order to obtain point cloud data with accurate geographical references), and base station (Galaxy 1). In both years, the parameter settings were the same. Six routes with different flight angles were merged to increase point density and reduce the sparsity of point clouds.

**Table 3.** Main technical parameters of the Riegl VUX-1 sensor.

| Parameters | Specification |
|---|---|
| Flying height | 15 m |
| Flight speed | 3 m/s |
| Scanning frequency | 550 kHz |
| Field of view angle | 330° |
| Scanning accuracy | ±10 mm |
| Scan speed | up to 200 scans/s |

### 2.2.3. In Situ Measurement Data

The data were measured within three days of the LiDAR data collection. Two plants were randomly selected from each plot for destructive sampling and transferred to the laboratory to measure PH and EH. Maize plants with relatively uniform canopies were selected within each plot to ensure the rationality and typicality of the study. PH and EH were measured as the distance from the bottom to the top of the maize plant and the distance from the bottom of the maize plant to the bottom of the first maize ear position, respectively [42] (Figure 2).

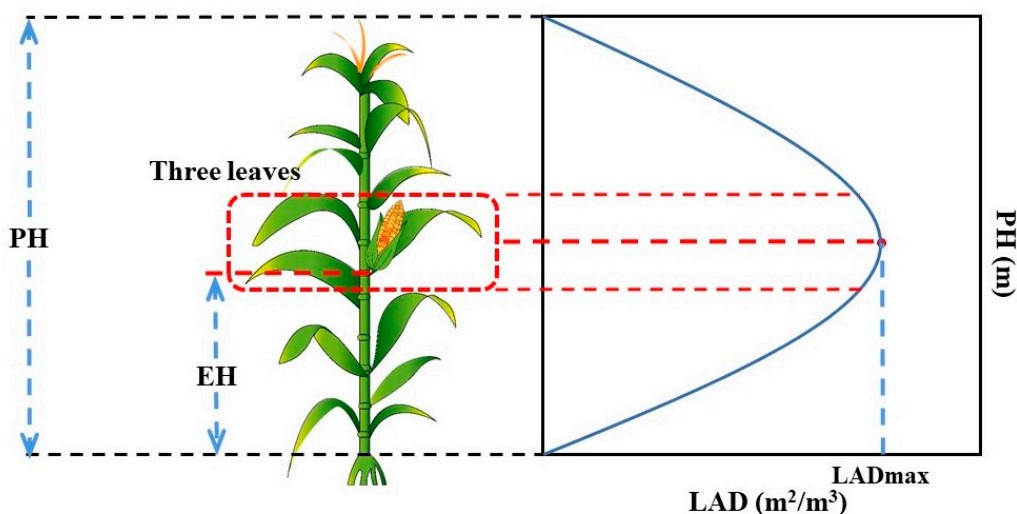

**Figure 2.** Schematic diagrams of a plant of maize and of LAD distribution with height2.3. LiDAR Data Preprocessing.

Raw data were stitched using the FARO SCENE software (version 2021.0) to generate TLS data. FARO SCENE can process and manage scanned data efficiently and easily by using automatic object recognition, scan stitching, and localization. The DLS data required calculation and data registration of different routes. The calculation was divided into two parts: trajectory calculation and raw laser data processing. First, using GPS base station data and position and orientation system (POS) data, accurate trajectory data are calculated by differential positioning algorithms [43] and the GPS level-arm inverse calculation of the POSPAC UAV software (version 7.1, Applanix Co., Richmond Hill, Canada) and then the RiProcess software (version 1.7.2, RIEGL Co., Southport, Queensland, Australia) to process the original laser data, performing waveform calculation, trajectory data matching, and 3D point cloud visualization. Finally, the data were exported as point cloud data in las format. The CloudCompare software (Open Source Project, version 2.10, www.cloudcompare.org, accessed on 26 June 2022.) was used to complete preprocessing for data from different platforms, including point cloud denoising, ground point filtering, and segmenting plots [35] (Figure 3).

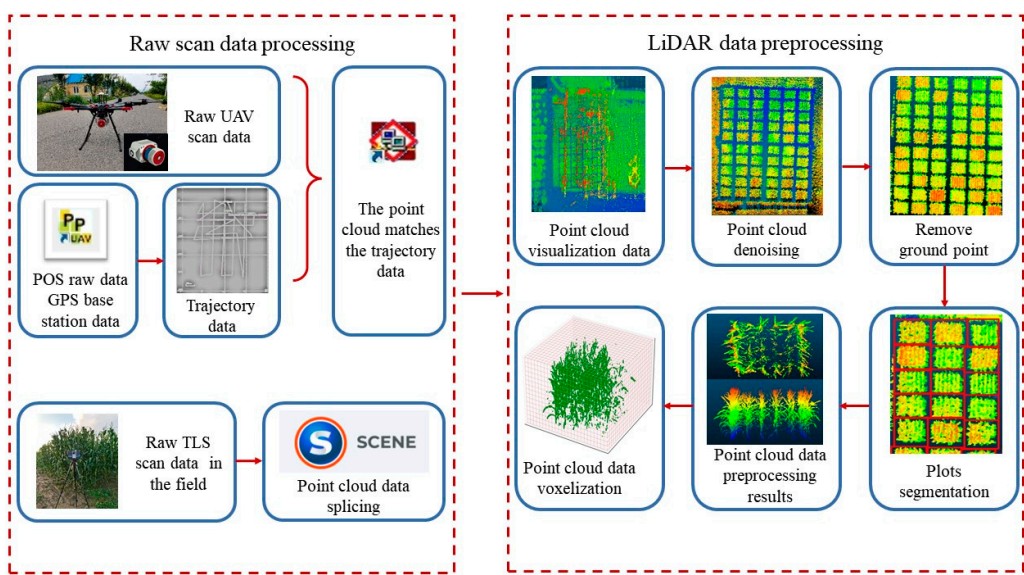

**Figure 3.** LiDAR data processing.

### 2.3. Point Cloud Data Voxelization

Voxels are defined as volume elements in a three-dimensional array, and voxelization of point cloud data is a commonly used method for processing point cloud data and grouping them into individual voxels. Different voxel sizes may have different effects on our experiments.

The boundaries of the voxel coordinates are determined by the minimum and maximum values of the *X*, *Y*, and *Z* Cartesian coordinates of the point cloud region. The main steps of voxelization are as follows. First, according to the boundary of the point cloud data, the point cloud is divided into $i \times j \times z$ voxels of size $\Delta i \times \Delta j \times \Delta z$ using Formula (1), where ($i$, $j$, $z$) represents the voxel coordinates in the voxel array, and *int* is an integer operator. Then, if there are laser points in the voxel, the voxel attribute containing the laser point is 1, and the voxel attribute that does not contain the laser point is 0 (i.e., the pores in the vegetation canopy).

$$\begin{cases} i = int\left(\frac{x_{max} - x_{min}}{\Delta i}\right) \\ j = int\left(\frac{y_{max} - y_{min}}{\Delta j}\right) \\ z = int\left(\frac{z_{max} - z_{min}}{\Delta z}\right) \end{cases} \tag{1}$$

### 2.4. LAD Estimation Model

The LAD is the sum of the leaf area on one side of the leaf per unit volume and its spatial distribution is called the LAD distribution. An important reason for determining the accurate EH estimates is to ensure that a point cloud can effectively reflect the vertical leaf area distribution. The experiment adopted the voxel method based on the contact frequency to estimate the LAD, which is highly sensitive to voxel size and can be affected by many factors [44]. LAD is calculated [45,46] as follows:

$$LAD(h, \Delta H) = \alpha(\theta) \times \frac{1}{\Delta H} \sum_{k=m_h}^{m_h + \Delta H} \frac{n_I(k)}{n_I(k) + n_P(k)} \tag{2}$$

where $n_I(k)$ and $n_P(k)$ denote the number of voxels with the attribute values of 1 and 0 in the k-th horizontal layer of the voxel array; $n_I(k) + n_p(k)$ is the total number of incident laser beams that reach the k-th layer; $\Delta H$ is the horizontal layer thickness of the leaf area density distribution. The smaller the $\Delta H$, the smaller the error of the estimated EH of the maize plot, so we set it as the voxel size; $\theta$ is the incident angle of the laser beam (i.e., the

angle between the direction of the laser pulse emitted by the LiDAR and the direction of the zenith); $\alpha(\theta)$ is the correction factor between the leaf inclination and the direction of the laser beam and, usually, has a value of 1.1 [35]; the LAD unit is usually $m^2/m^3$.

### 2.5. Estimation Model of Maize EH and ER

Following agronomic principles, the area of the ear leaf is the largest, so the LAD is greatest at the ear position. Based on the relationship between LAD and PH [41], the average EH of plot-scale maize is the height at which LAD peaks (Figure 2). However, this is not the same as the EH we measured; therefore, we introduced the constant C to eliminate this error. To reduce the influence of noise in the point cloud data, we added a 95% confidence interval to remove the top 5% of the point cloud to ensure that the EH is less affected by noise. Most of the current extraction methods of PH are based on the determination of the crop surface model (CSM) and the digital terrain model (DTM). The difference between the two models is the PH. We calculated PH using the difference between the mean value of the upper 5% of the point cloud data and the mean value of the lower 5% of the point cloud data to increase its representativeness. The EH and ER estimation formulas are as follows:

$$EH = \frac{L_{(LAD(max))} \times H_{max} \times 95\%}{L} - \frac{\Delta H}{2} - C \tag{3}$$

$$ER = \frac{EH}{PH} \tag{4}$$

where $L_{(LAD(max))}$ is the voxel layer number where the LAD is the largest; $H_{max}$ is the maximum PH of maize in this plot; $L$ is the total number of layers of maize in the plot, i.e., it is the value of $H_{max}/\Delta H$ within each plot. $\Delta H$ takes the same size as the voxel size, so it can also be considered as the number of voxel layers; $C$ is a constant. It is the difference between the actual definition of EH and the definition of EH through LAD, which is half the length of the stem node. In this paper, the value of $C$ was set to 10 cm.

### 2.6. Accuracy Evaluation

This paper determined the optimal voxel size for estimating the EH of maize based on LiDAR point cloud data from different platforms, taking into account that each instrument has a different bias. We used the coefficient of determination ($R^2$) and the root mean square error (RMSE) for accuracy verification. Higher model accuracy was indicated by an $R^2$ closer to 1 and lower RMSE. The $R^2$ and RMSE were calculated as follows:

$$R^2 = 1 - \frac{\sum_{i=1}^{n} (y_i' - \overline{y})^2}{\sum_{i=1}^{n} (y_i - \overline{y})^2} \tag{5}$$

$$RMSE = \sqrt{\frac{\sum_{i=1}^{n} (y_i - y_i')^2}{n}} \tag{6}$$

where $y_i$ is the measured EH value of plot i; $\overline{y}$ is the average of the measured EH values of each plot; $y_i'$ is the EH value of plot $i$ derived from the point cloud; $n$ is the number of plots, and two years of data are used in this study; $n = 60$ and $n = 68$, respectively.

## 3. Results

### 3.1. Optimal Voxel Selection for Different Platforms

Optimal Voxel Selection for TLS Platform

The voxel size is an important parameter in this method; because of LAD's high sensitivity to voxel size [33] and its effect on the estimation of EH, it is important to determine the optimal voxel. The distribution curves of LAD varied for different voxel sizes, as can be noticed for the mean values of all plots of each cultivar for two years (Figure 4). As the voxel increases, the LAD of each layer also becomes larger, but the

general shape of the LAD distribution curve remains unchanged. In addition, we found that the LAD distribution curves generally had only one or two peaks (A1 had only one peak, while all the remaining cultivars had two peaks), indicating that the curves may have varied in the different plant types and leaf distributions of the different cultivars.

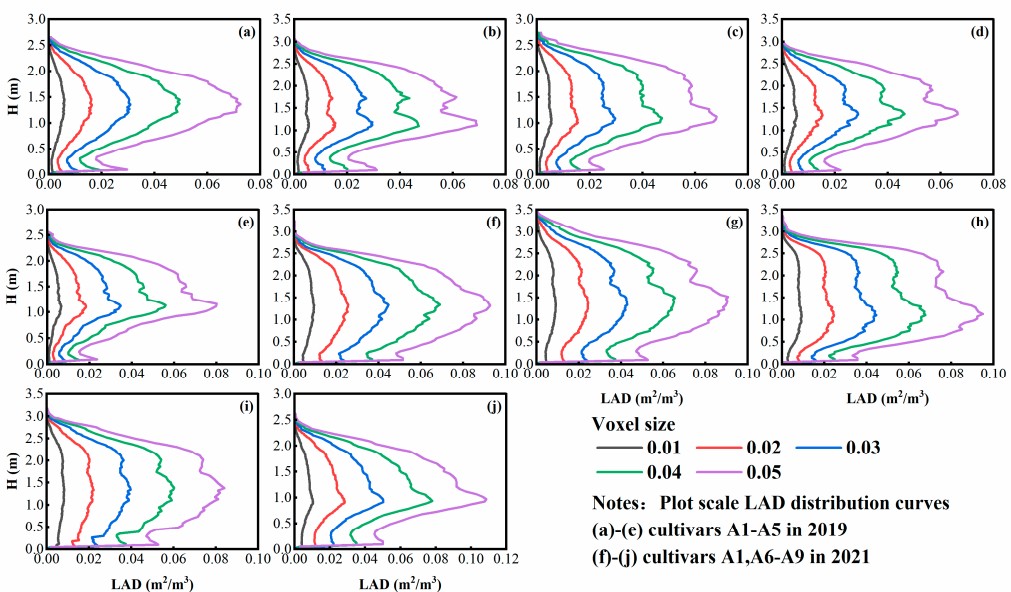

**Figure 4.** Plot-scale LAD distribution based on TLS under different voxel size at the filling period (**a**–**e**) cultivars A1–A5 in 2019 and (**f**–**j**) cultivars A1, A6–A9 in 2021.

Thus, the effect of voxel size on EH estimates was analyzed using maize plots of all planting densities. We used the 2019 plot-scale data for voxel size selection, showing the highest accuracy of $R^2 = 0.59$ and RMSE = 16.90 cm at a voxel size of 2.00 cm (Figure 5a). We also validated the results using the 2021 data, showing the highest accuracy of EH estimation at $R^2 = 0.39$ and RMSE = 18.40 cm for the same voxel size (Figure 5b). For the TLS platform, the average point distances of the data acquired in both years were 0.13 cm and 0.17 cm, respectively; thus, the optimal voxel size was 11.76–15.38 times the optimal point distance.

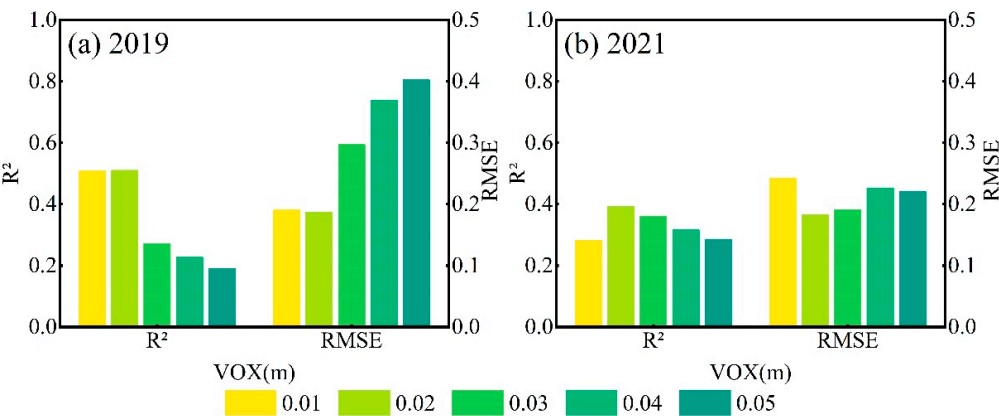

**Figure 5.** Plot-scale optimal voxel selection for TLS data (**a**) in 2019 and (**b**) in 2023.1.2. Optimal Voxel Selection for DLS Platform.

Considering the different features of laser beam diameter, laser scanner distance, and resolution of the different platforms, we revisited the optimal voxel size for the DLS platform. Our voxel sizes were chosen to be 2.00 cm–10.0 cm with an interval of 2.00 cm.

Unlike the results for the TLS platform (Figure 5), for the 2019 DLS platform data, the best EH estimates had $R^2 = 0.59$ and RMSE = 18.00 cm when the voxel size was 10.00 cm (Figure 6a). We used data from the 2021 DLS platform for validation, and the results were consistent with those of 2019, with the best EH estimate of $R^2 = 0.46$ and RMSE = 26.50 cm at a voxel size of 10.00 cm (Figure 6b). For the DLS platform, the average point distance of the data acquired in both years is 1.98 cm, so the optimal voxel size was 5.05 times the optimal point distance. Obvious overestimation occurs when the density is greater than D3, so we only used data with densities at or below D3. Since commercial cultivars were used in our experiments, stable agronomical performance can be expected; nearly no case exceeds 1.6 m for the EH according to the declared cultivar's plantation instruction. Based on this a priori knowledge, some plots with severe overestimation are outliers, and we remove these outliers. The results show that the optimal voxel size differs among sensors. However, it should be pointed out that the threshold of 1.6 m is not necessarily fixed and may be invalid for a few cases (e.g., inbred lines' material has more complex performance and their growth performance is unpredictable) and needs to be adjusted based on a priori knowledge of the actual scenario.

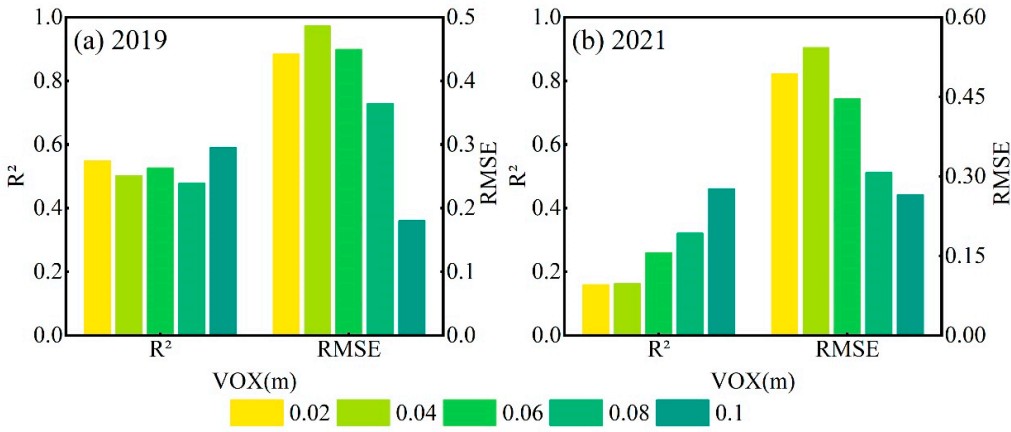

**Figure 6.** Plot-scale optimal voxel selection for DLS data (**a**) in 2019 and (**b**) in 2021.

*3.2. Comparison of EH and ER Estimation for Different Planting Densities*

We used a one-way ANOVA approach for the final experiment to compare the estimation results from different platforms. The DLS platform was used only for plots with overestimated planting density of D3 and below. The results showed significant differences in EH estimation between cultivars, which demonstrated the reproducibility of our method (Table 4).

**Table 4.** One-way analysis of variance (ANOVA) for EH between cultivars.

| Year | Source | SS | df | MS | F | P > F |
|------|--------|-----|-----|------|------|-------|
| 2019TLS | Inter-category | 0.743 | 4 | 0.186 | 12.861 | 0.000 |
| | Intra-category | 0.794 | 55 | 0.014 | — | — |
| | Total | 1.537 | 59 | — | — | — |
| 2021TLS | Inter-category | 1.066 | 4 | 0.266 | 11.448 | 0.000 |
| | Intra-category | 1.466 | 63 | 0.023 | — | — |
| | Total | 2.532 | 67 | — | — | — |
| 2019DLS | Inter-category | 0.325 | 4 | 0.081 | 3.478 | 0.026 |
| | Intra-category | 0.467 | 20 | 0.023 | — | — |
| | Total | 0.791 | 24 | — | — | — |
| 2019DLS | Inter-category | 0.935 | 4 | 0.234 | 8.847 | 0.000 |
| | Intra-category | 1.163 | 44 | 0.026 | — | — |
| | Total | 2.098 | 48 | — | — | — |

### 3.2.1. Comparison of EH and ER Estimation for Different Planting Densities for the TLS Platform

In 2019, when the planting density was of D4 or less, the $R^2$ values of the estimated EH were both around 0.60, and the RMSE was around 15.00 cm (Table 5). At the planting density of D5, the $R^2$ was higher, but RMSE also increased significantly to 21.60 cm. Figure 7a shows that for 85% of the plots, EH was overestimated in 2019. At planting densities of D2, D3, D4, and D5, EH was overestimated by more than 20 cm at 13.33%, 13.33%, 20.00%, and 53.33%, respectively. Thus, the overestimation of EH was more severe when the planting density was too high and the RMSE increased with increasing planting density.

**Table 5.** $R^2$ and RMSE for different densities of EH with TLS data in 2019 and 2021.

| Year | | D1 | D2 | D3 | D4 | D5 | Total |
|------|------|------|------|------|------|------|------|
| 2019 | $R^2$ | — | 0.60 | 0.55 | 0.65 | 0.68 | 0.59 |
| | RMSE (cm) | — | 14.60 | 15.60 | 15.70 | 21.60 | 16.90 |
| 2021 | $R^2$ | 0.38 | 0.44 | 0.49 | 0.40 | — | 0.39 |
| | RMSE (cm) | 17.20 | 20.80 | 15.30 | 18.50 | — | 18.40 |

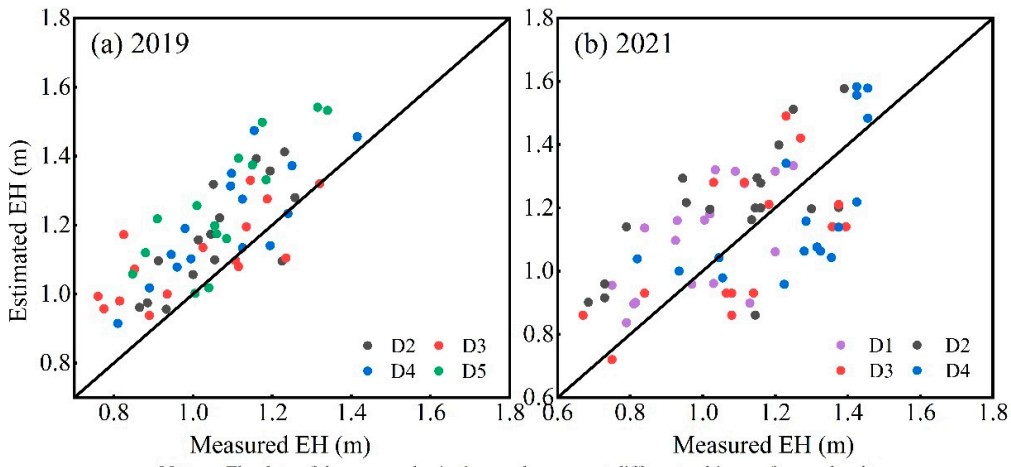

Notes: The dots of the same color in the graph represent different cultivars of same density.

**Figure 7.** Plot-scale filling period EH estimates at different planting densities from TLS data (**a**) in 2019 and (**b**) in 2021.

In 2021, the estimated EH had an $R^2$ = 0.39 and RMSE = 18.40 cm and had better results for $R^2$ and RMSE at all planting densities. However, $R^2$ was lower and RMSE was higher than in 2019 at all planting densities. Likely this was due to a large amount of hail-damaged maize leaves in 2021. When planting densities were D1 and D2, the estimates were inferior than with densities D3 and D4. This may have been due to the low density of D1 and D2, which was caused by the hail that collapsed some maize plants within the same plot. However, at D3 and D4 planting densities, the results were still overestimated. Therefore, at higher planting densities, there is a more serious overestimation of EH at the plot scale. Accurate estimation of the PH is a key step in obtaining ER. The PH estimated from the 2019 TLS data had $R^2$ = 0.86 and RMSE = 9.90 cm (Figure 8), providing the basis for estimating ER. Based on the results of EH and PH, we obtained the ER results for 2019. The ER did not show a decrease with increasing planting density; instead, the $R^2$ was greatest at a density of D5 (Table 6). Overall, the $R^2$ was 0.45 and the RMSE was 0.06. The ER estimates for different planting densities also all have an $R^2$ around 0.40 and RMSE around 0.06. Therefore, the relationship between ER estimates and planting density was not significant, probably because ER is the result of the combined effect of PH and EH.

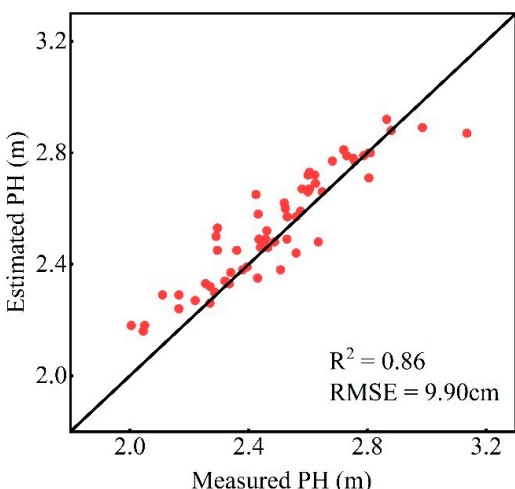

**Figure 8.** Filling period PH estimates from 2019 TLS data.

**Table 6.** $R^2$ and RMSE for different densities of ER with TLS data in 2019.

| Year | | D1 | D2 | D3 | D4 | D5 | Total |
|------|------|------|------|------|------|------|-------|
| 2019 | $R^2$ | — | 0.48 | 0.35 | 0.34 | 0.64 | 0.45 |
|      | RMSE (cm) | — | 0.06 | 0.06 | 0.06 | 0.06 | 0.06 |

### 3.2.2. Comparison of EH and ER Estimation for Different Planting Densities for the DLS Platform

The applicability of our method on this platform was analyzed using DLS data collected in 2019 and 2021 to acquire EH. Applying the EH estimation model to DLS data at current planting densities led to greater overestimations (Figure 9). We counted the ratio of abnormal EH under the same planting density. In 2019, at planting densities D2, D3, D4, and D5, the EH estimated as outliers accounted for 6.67%, 33.33%, 26.67%, and 33.33%; the estimated EH greater than 30 cm accounted for 6.67%, 33.33%, 53.33%, and 66.67%. Table 7 also shows that when the density was greater than D3, $R^2$ decreased and RMSE increased sharply. When the planting density exceeded D3, there was a serious model overestimation. Based on a priori knowledge, we removed the outliers with planting density D3 and below. After removal, the EH estimates for planting densities at D3 and below showed $R^2 = 0.59$ and RMSE = 18.0 cm. The results are consistent with the TLS data; higher planting density led to worse estimation accuracy and larger errors.

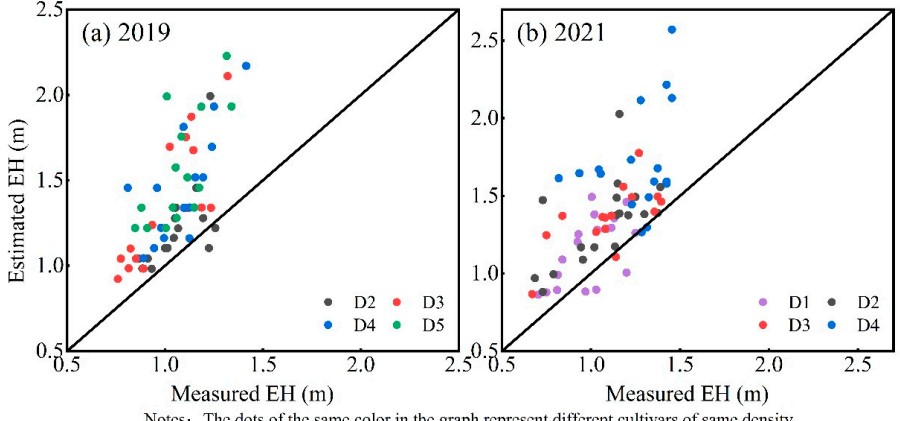

Notes: The dots of the same color in the graph represent different cultivars of same density.

**Figure 9.** EH estimation at different planting densities at plot scale for filling period for DLS data (**a**) in 2019 and (**b**) in 2021.

**Table 7.** $R^2$ and RMSE of EH of different densities from DLS data in 2019 and 2021.

| Year | | D1 | D2 | D3 | D4 | D5 | Total |
|---|---|---|---|---|---|---|---|
| 2019 | $R^2$ | — | 0.40 | 0.66 | 0.54 | 0.48 | 0.49 |
| | RMSE (cm) | — | 25.00 | 42.50 | 43.90 | 53.70 | 42.60 |
| 2021 | $R^2$ | 0.37 | 0.37 | 0.46 | 0.10 | — | 0.41 |
| | RMSE (cm) | 23.60 | 33.90 | 30.40 | 56.50 | — | 38.10 |

The 2021 estimated EH data were consistent with those of 2019. The majority of the plots (58.82%) had outliers for EH estimates when the planting density exceeded D4. At planting densities D1, D2, and D3, the proportion of plots with EH estimated as outliers was zero, 5.67%, and 6.67%, respectively (Figure 9b). At densities of D3 and below, the $R^2$ was 0.46 and RMSE was 26.50 cm.

The PH values estimated for the 2019 DLS platform were analyzed, showing $R^2 = 0.8$ and RMSE = 11.10 cm (Figure 10). We estimated the ER using the same data as for the EH and the PH estimated from the DLS data. The ER results of D3 and D2 are shown below (Table 8). In total, the ER estimates were $R^2 = 0.41$ and RMSE = 0.08. The $R^2$ and RMSE of each planting density also had better results.

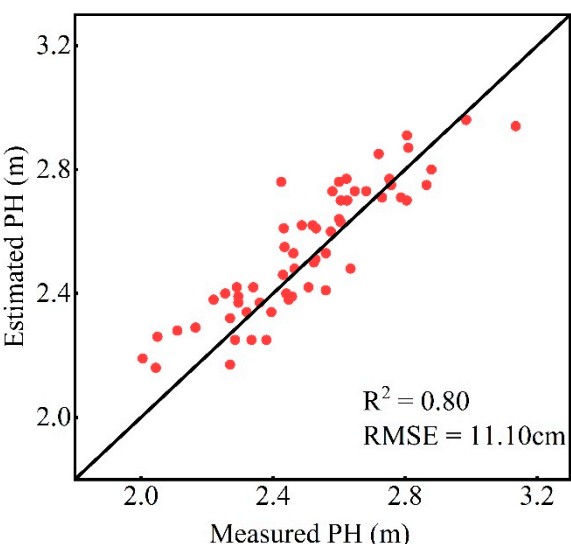

**Figure 10.** Filling period PH estimates from DLS data in 2019.

**Table 8.** $R^2$ and RMSE of ER of different densities on DLS platform in 2019 and 2021.

| Year | | D1 | D2 | D3 | Total |
|---|---|---|---|---|---|
| 2019 | $R^2$ | — | 0.37 | 0.55 | 0.41 |
| | RMSE (cm) | — | 0.07 | 0.08 | 0.08 |

### 3.3. Comparison of EH and ER Estimation under Different Cultivars of TLS Platform

The proposed model relies on the maximum LAD of the three leaves at the ear position, but different cultivars do not necessarily have the maximum leaf area of the ear position leaves [47]. Li et al. [39] pointed out that maize generally has 22 leaves at the mature stage, and the ear position leaf of A2 is generally the 14th, which has the largest leaf area; the ear position leaf of A1 is generally the 16th leaf, which does not have the largest leaf area. Only data from 2021 were used because the 2019 measurements did not mark the ear position leaves. Figure 11 shows the average leaf area of each leaf measured for 32 maize plants of the same cultivar, where $LA_{(max)}$ is the leaf area of the largest leaf, $LA_{(ear)}$ is the leaf area of the ear position leaf, and $LA_{(max\ and\ ear)}$ is the leaf area of the ear position leaf and the

largest leaf when they are the same leaf. The ear position leaves of A1 and A8 were located below the leaves with the largest leaf area. The ear position leaves of A6 and A7 had the largest leaf area. The ear position leaves of A9 were above the leaves with the largest leaf area. Therefore, the EH of different maize cultivars is not necessarily at the leaf with the largest area, but around the leaf with the largest leaf area.

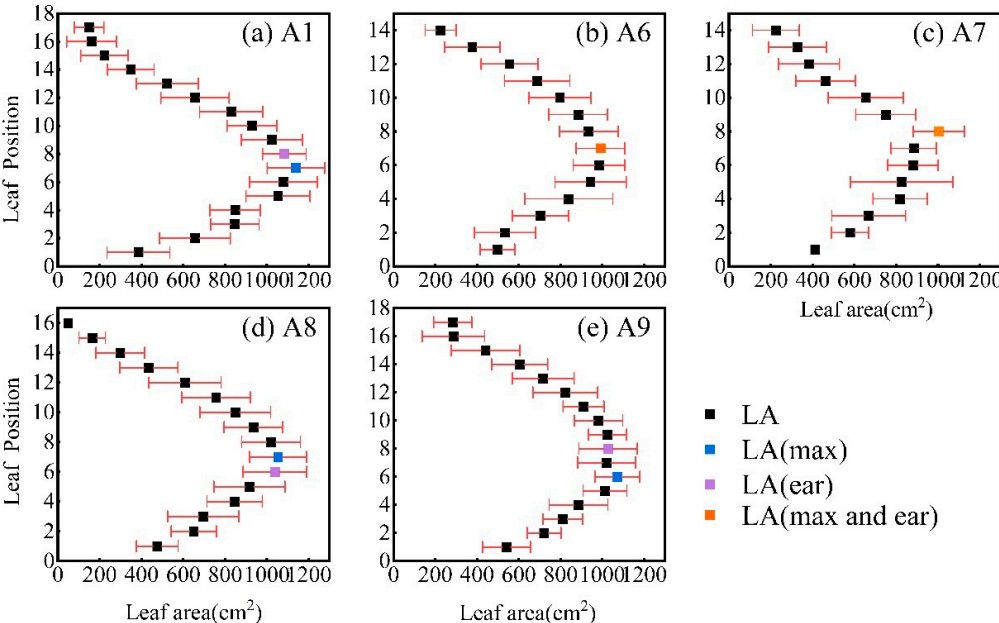

**Figure 11.** Relationship between EH and leaf area of different cultivars at filling period in 2021 (**a**–**e**) cultivars A1, A6–A9 in 2021.

Experiments in 2019 and 2021 used a total of nine maize cultivars, including all maize plant types (compact: A1, A2, A3, and A7; semi-compact: A4, A5, A6, and A8; scattered: A9). We used TLS data to analyze the effect of different plant types and cultivars on the estimation of EH. We counted the proportion of EH-overestimated plots of each species to their total number of plots and the proportion of plots with EH overestimated by over 20 cm to their total number of plots, but we did not detect a clear pattern (Table 9). Overestimation did not become more severe as plant types became more compact (Figure 12). Overestimation was also not related to the location of the largest leaf.

**Table 9.** EH overestimation using TLS data for different cultivars.

| Cultivar | Plots with Overestimated EH/Total Number of Plots of This Cultivar (In 2019) | Plots with EH Overestimated by More than 20 cm/Total Number of Plots of This Cultivar (In 2019) | Plots with Overestimated EH/Total Number of Plots of This Cultivar (In 2021) | Plots with EH Overestimated by More than 20 cm/Total Number of Plots of This Cultivar (In 2021) |
|---|---|---|---|---|
| A1 | 83.33% | 33.33% | 84.67% | 7.67% |
| A2 | 75.00% | 8.33% | — | — |
| A3 | 83.33% | 16.67% | — | — |
| A4 | 83.33% | 16.67% | — | — |
| A5 | 100.00% | 33.33% | — | — |
| A6 | — | — | 57.1% | 7.14% |
| A7 | — | — | 38.46% | 23.08% |
| A8 | — | — | 58.33% | 16.67% |
| A9 | — | — | 81.25% | 37.50% |

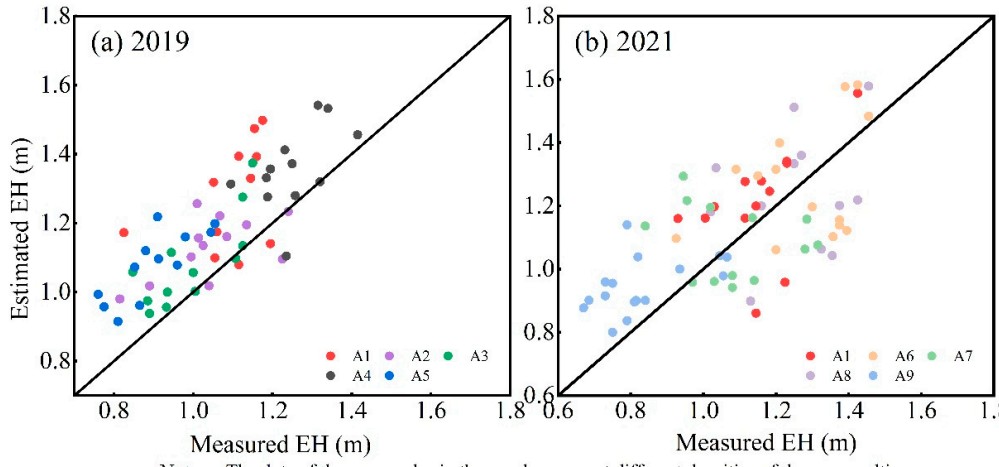

**Figure 12.** Filling period plot-scale EH estimates from TLS data for different cultivars (**a**) in 2019 and (**b**) in 2021.

The effect of different cultivars on ER estimation was analyzed using 2019 TLS data. The estimated ER did not correlate well with the cultivars (Figure 13). This may have been because the accuracy of ER estimation is influenced by various factors such as PH, EH, planting density, and cultivar, not only the cultivar. The model's EH and ER estimates for different maize cultivars were not significantly related to maize type as expected. However, the model had some stability across cultivars. Overall, the model still improved EH and ER estimation, indicating that the proposed model can be used for large-scale, high-throughput estimation of EH and ER for different cultivars.

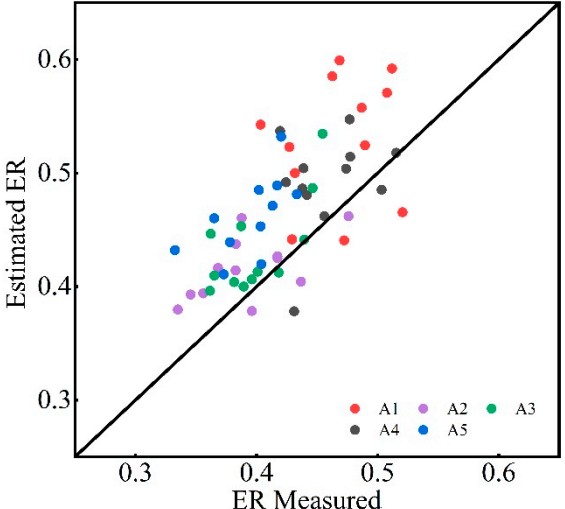

**Figure 13.** TLS data to estimate different cultivars at plot-scale ER at the filling period in 2019.

## 4. Discussion

### 4.1. Advantages of the EH Estimation Model Compared to Similar Studies

Several studies have used machine vision methods to explore the possibility of maize ear position identification. Brichet et al. [17] developed a repeatable pipeline based on ear detection algorithms to track the ear growth of hundreds of plants. In their case, 86.00% of maize ears were accurately identified. Jia et al. [48] combined deep learning with image processing to develop a multiscale hierarchical algorithm to identify maize ears, and the method achieved 97.02% recognition accuracy. However, the machine-vision-based methods all identified ear positions directly, which does not overcome ear position

occlusions caused by high planting densities in breeding fields. In addition, relying on photography for identification requires the consideration of weather conditions and time of acquisition to ensure photo quality. Most of the existing studies only identify corn ear positions, but do not measure EH.

Compared with previous methods, our proposed model accomplishes the extraction of the EH. LiDAR data and agronomic knowledge are combined to obtain EH not by directly identifying the maize ear but by ear position determined by the leaves. This means that we do not need a complex algorithm to determine ear position and are less affected by errors due to the shading of the ear. LiDAR provides an efficient way to collect data and is less affected by weather and environment conditions than other instruments. The proposed model provides an approach for large-scale acquisitions of EH.

### 4.2. Uncertainty in Fitting the LAD Distribution Curve

Model-fitting methods can effectively suppress the noise of data. In addition, they are expected to be more objective and easier to adapt to a wide range of situations [49]. Cubic spline interpolation is known to resample discrete data based on the least squares method with the cubic convolution interpolation function [50]. We used cubic spline interpolation to fit discrete LAD and LAD vertical distribution scatter plot to obtain a smooth LAD vertical distribution curve that could be used to estimate the average EH at the plot scale. We normalized the PH before fitting (Formula (7)). The EH at the plot scale was estimated according to the smoothed curve (Figure 14).

$$H_{norm} = \frac{H_i - H_{min}}{H_{max} - H_{min}} \tag{7}$$

where $H_{norm}$ is the normalized PH, $H_i$ is the PH estimated by the point cloud, $H_{min}$ is the smallest PH in the plot, and $H_{max}$ is the largest PH in the plot.

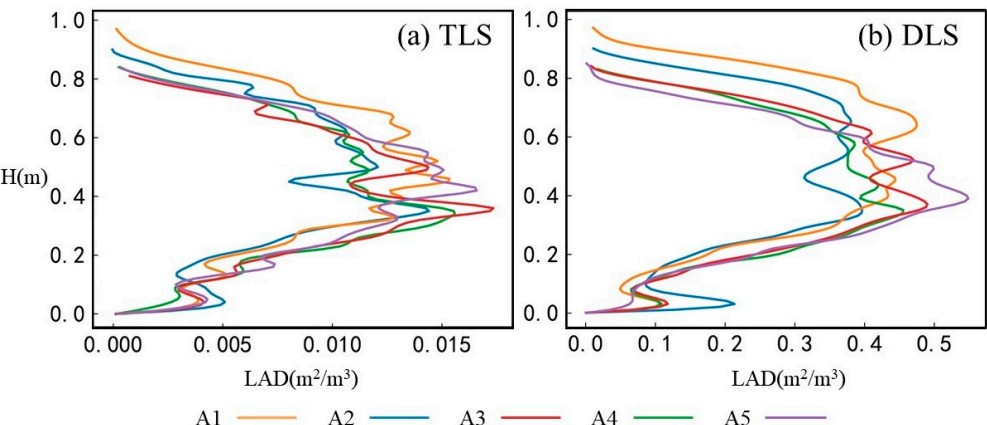

**Figure 14.** Distribution of LAD curves after curve fitting for different cultivars at the filling period in 2019 for (**a**) TLS data and (**b**) DLS data.

The LAD distribution curve was interpolated at equal intervals (0.01 m) using cubic spline interpolation. The LAD distribution curve after interpolation fitting was smoother, reducing the influence of noise points, but this had little effect on the results of the average EH at the plot scale estimated by each sensor (Tables 10 and 11) and did not significantly improve (Tables 5 and 7). For the TLS data, the EH of one plot has an outlier in 2019 and the EH values of two plots have outliers in 2021. For DLS data, the overestimation after fitting is not significantly improved.

**Table 10.** $R^2$ and RMSE after curve fitting of data on different platforms in 2019.

| Platform | | D2 | D3 | D4 | D5 | Total |
|---|---|---|---|---|---|---|
| 2019 DLS Curve Fitting | $R^2$ | 0.44 | 0.62 | 0.56 | 0.49 | 0.49 |
| | RMSE (cm) | 22.70 | 39.10 | 39.20 | 50.60 | 39.20 |
| 2019 TLS Curve Fitting | $R^2$ | 0.58 | 0.48 | 0.52 | 0.71 | 0.54 |
| | RMSE (cm) | 15.50 | 17.70 | 18.50 | 21.60 | 18.40 |

**Table 11.** $R^2$ and RMSE after curve fitting of data on different platforms in 2021.

| Platform | | D1 | D2 | D3 | D4 | Total |
|---|---|---|---|---|---|---|
| 2021 DLS Curve Fitting | $R^2$ | 0.44 | 0.34 | 0.46 | 0.10 | 0.41 |
| | RMSE (cm) | 22.70 | 36.80 | 30.30 | 51.70 | 38.10 |
| 2021 TLS Curve Fitting | $R^2$ | 0.40 | 0.71 | 0.41 | 0.38 | 0.43 |
| | RMSE (cm) | 17.20 | 19.10 | 18.20 | 24.00 | 18.60 |

In this study, the cubic spline interpolation function was used to fit the LAD distribution curve; it did not result in significant LAD improvements. It is worth discussing whether this method can effectively reduce the influence of noise points. Various platforms and sensors acquire data in different ways. It is expectable that using different fitting methods in future studies would be beneficial.

### 4.3. Comparison of Different Data Collection Methods

We used different platforms and sensors to acquire data in this study. In general, TLS platforms scan from the middle layer, and this scanning approach makes the middle layer less affected by the canopy, and the data in the middle layer can be acquired more completely. The DLS platform takes a top-down scanning approach. Occlusion effects are caused by leaves intercepting the laser beams and preventing them from coming in contact with the material along their path [32]. That is, from the perspective of the DLS platform, the top layer contains the most information, while the middle and bottom layers have some obscured leaves, resulting in missing data [51]. Therefore, the top-down scanning approach of the DLS platform will make the middle and bottom layers contain less complete information than the TLS platform. As such, TLS-derived structural and functional traits have higher accuracy than DLS-derived traits [52]. This leads to a significant overestimation of the DLS platform compared to the TLS platform when estimating EH.

### 4.4. The Influence of Vertical Distribution of Point Density and Missing Point Cloud

The different LiDAR systems have different specifications and setup configurations, resulting in different locations of canopy elements not being sampled equally [53]. This results in different average point distances on each vertical profile, which affects the point density distribution. The point density distribution may be one factor affecting the LAD estimation. Therefore, we counted the average point distance across vertical profiles to determine the effect of point density on our proposed method. We counted the average point distances on the vertical profiles of the five plots at D5 densities estimated for EH overestimation using DLS data in 2019. We determined the effect of the vertical point density distribution on our model. With the layer height H set to 2.00 cm, the average point distances are basically the same in most of the middle areas (Figure 15). When the voxel size is set to 10.00 cm, it is larger than the average point distance, indicating that the vertical point density distribution is not the cause of the overestimation. Although the vertical point density distribution is consistent, it does not mean that the problem of canopy shading is solved, and missing data are still inevitable.

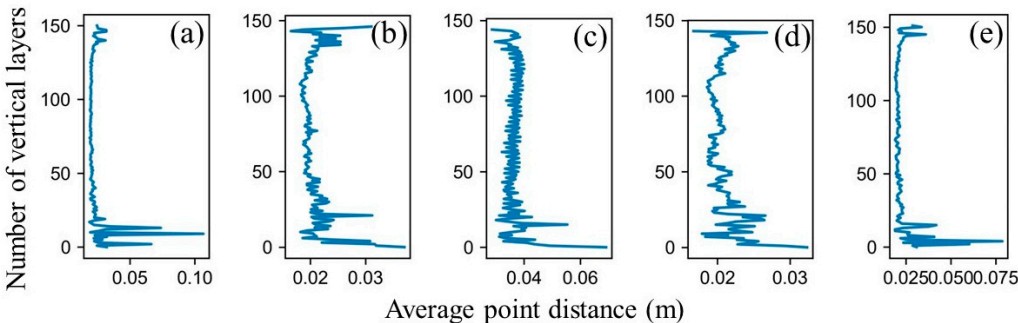

**Figure 15.** Vertical point distance distribution of plots with the DLS platform. (**a**–**e**) are the random five plots of EH overestimation at D5 planting density with cultivars A1-A5, respectively.

When the planting density is too high, the leaves will overlap and canopy shading becomes severe, resulting in missing point cloud data in the middle and lower layers. Figure 16a,b shows the maize point cloud data and LAD distribution curves at D2 planting density, and Figure 16c,d shows the maize point cloud data and LAD distribution curves at D5 planting density. It is easy to see that the point cloud in the middle and lower layers is seriously missing at high planting densities, resulting in EH overestimations. Therefore, our model cannot solve the EH estimation of breeding fields with too high a planting density for the time being.

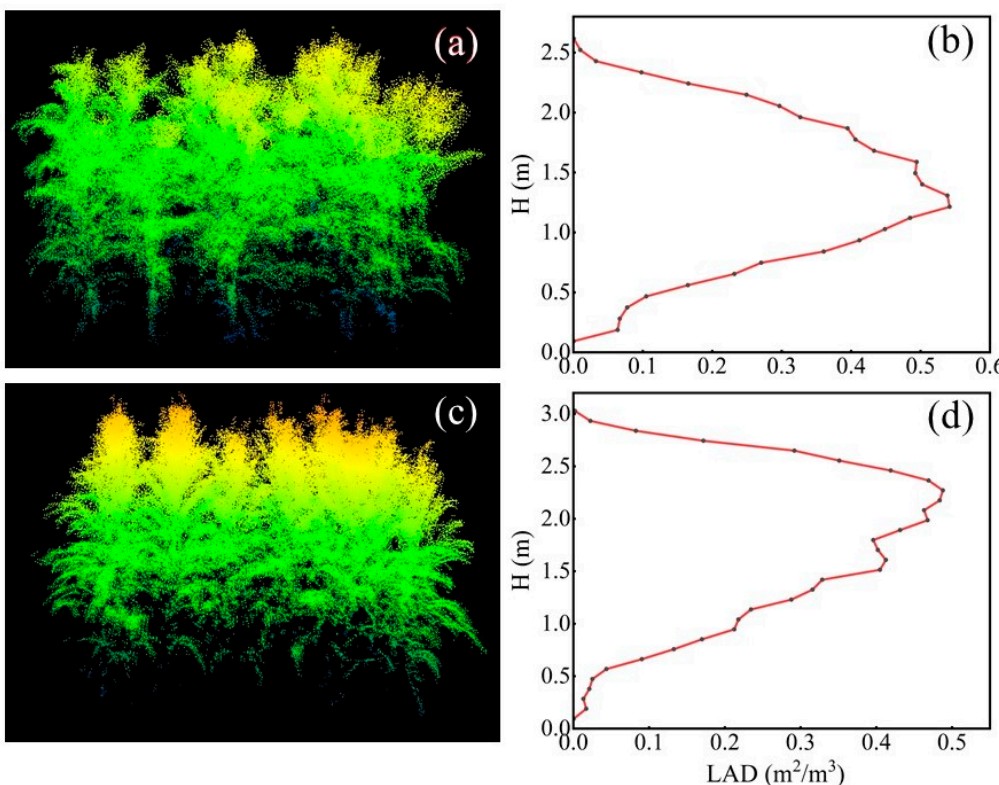

**Figure 16.** Missing point clouds for different planting densities in the DLS platform. (**a**,**b**) are the maize point cloud data and LAD distribution curves at D2; (**c**,**d**) are the maize point cloud data and LAD distribution curves at D5.

## 5. Conclusions

This paper proposed a novel and simple estimation model of the maize EH and ER based on LiDAR data, demonstrating the model's application on different platforms, densities, and cultivars.

The following conclusions were obtained. (1) The EH estimation model proposed in this paper can measure EH relatively accurately for different planting densities and cultivars of maize. The $R^2$ was greater than 0.5, and the RMSE was about 20.0 cm. (2) This model currently works best with the TLS platform. When the voxel size was 2.00 cm, the overall accuracy in 2019 reached $R^2 = 0.59$ and RMSE = 16.90 cm. When the voxel size was 10.00 cm, the model could also be used for DLS platforms, but only for smaller planting densities. In 2019, when the planting density was 67,500 plants/ha and below, $R^2 = 0.59$ and RMSE = 18.00 cm. (3) We proposed an ER estimation method with $R^2 = 0.45$ and RMSE = 0.06 when using TLS data, which can estimate field maize ER with relative accuracy.

The EH and ER estimation model proposed in this paper somewhat improves the current time-consuming estimations of EH and ER of maize, but there are still problems in the estimation accuracy of high-density maize. In future work, efforts should be made in inbred lines' experimental fields to refine and explore this method and further determine its applicability to breeding programs. In addition, further exploration on addressing canopy occlusion and leaf area density vertical distribution extraction accuracy needs to be carried out. This method shows promise in utilizing multi-angle data from the DLS platform and increasing the penetration of the LiDAR sensor to reduce the effects of canopy shading and missing point clouds. We expect that this approach will provide a method for efficient and portable high-density maize EH and ER detection.

**Author Contributions:** H.W. and L.L. conceived this study. H.W., W.X., C.Z. and R.C. performed experiments and data collection; S.H., G.Y., W.Z. and H.Y. helped to revise the manuscript. All authors participated in this study's implementation and completion. All authors have read and agreed to the published version of the manuscript.

**Funding:** This research was funded by the National Key Research and Development Program of China (2021YFD2000102), National Natural Science Foundation (42161059), and National Natural Science Foundation (32160365).

**Data Availability Statement:** Data sharing not applicable.

**Acknowledgments:** The authors would like thank Bo Xu for acquiring data in the field experiments of this study.

**Conflicts of Interest:** The authors declare no conflict of interest.

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
