# Peer review of "Maize Ear Height and Ear–Plant Height Ratio Estimation with LiDAR Data and Vertical Leaf Area Profile"

_remotesensing, doi:10.3390/rs15040964_

Round 1

Reviewer 1 Report

The authors present a novel study to estimate maize ear height (EH) and ear-plant height ratio (ER) using two LiDAR platforms: ground based Terrestrial Laser Scaner (TLS) and Drone Laser Scanner (DLS).

The manuscript is scientically sound an the experiment design (train/test, replications) adequated. The reproducibility of the results is uncertain, given the low R2 scores (below 0.5 on the test set), high RMSE (above 18 cm)  and the fact that a fixed constant value (10 cm offset) was added on Equation 3 to make it work with the training data. However, the work is novel, reproducibility requieres further research a big data, so at this stage it is acceptable given its novelty.

Table 4 is not usefult since it does not say how many measurement samples were taken, it only reports some basic statistics (mean, max, min, CV), but they are not compared at all with the results. In fact Equations 5 and 6 are not explained at all, one has to assume that samples were taken and compared to the estimates from LiDAR, what is yi y'i ? how many samples is more important than the data shown in Table 4.

Figure 11 is supposedly showing LA, LA(max), LA(ear), LA(max and ear) with different colours, however, in the images only appear one, two or maximum three of these labels, do they supperpose? It is really unclear.

In the materials and method section there is a paragraph (lines 134-140) indicating that there are four Nitrogen fertilizer rates in the experimental field. However, the methods section does not use these fetilizer rates at all in the manuscript, it seems this treatment has nothing to do with the metodology, centered on LiDAR pre-processing and use to estimate  EH and ER, there is no comment at all on this in the rest of the paper, so comment on this or remove it since it is inconsequential.

Specific comments:

 lines 145-147 states that 5 stations were evenly distributed among the four plots (Fig. 1), I do not see 5 stations, I see 4-6 on each row and column.

line 152, why do you selected those dates specifically to take  the DLS data?

line 155, you state that there is an antena to acquire satellite signals, there are many, many satellite signals, specify which signals (GPS?) and why.

line 177, what is POS RAW data and we do not know (people from remote sensing but not specifically on LiDAR) what differential calculation you are talking about, better use a reference.

Figure 2 right shows a lot of pre-processing (point cloud denoising, remove point ground, plot segementation) but none of this pre-processing is explained (software used, references), the manuscript just jumps to Point cloud data voxelization. In addition, section 2.3 is Acquisition and preprocessing of UAVs images and has nothing written on this section, the next line corresponds to section 2.4 (is it 2.3.1?).

lines 219-220 is there a reference citing the relationship between LAD and PH? (Equation 3)

lines 297-298, it is missing some phrase before "of the estimated EH"

line 353 the results shown are for D2 and D3 not only D3, in fact lines 358-360 indicates why, these lines should be moved up to line 297 to explain why Table 8 only shows two densities.

Line 431, Equation 7, this equation is introduced in the text without explanation (what is Hi, Hmin, Hmax) it is also wrong since Hi on the left hand of the Equation cannot be equal to the right hand of the equation and have no relationship whatsoever this the text before and after the equation (what is the index i anyway?)

Figure 15 does not indicates what are (a), (b), (c), (d), and (e) in the figure.      

Reviewer 2 Report

Reviewers Comments

This experiment presented a creative method to predict maize ear height (EH) using point cloud data combined with agronomic knowledge. The data are sound and the approach used is highly innovative. The work will provide an efficient and fast method of measuring EH for breeding programs. Before accepting for publication, several concerns should be addressed as follows:

 Line 73: In ‘the existing crop PH detection model is unsuitable for detecting EH.’, it would be more appropriately to be replaced with ‘existing crop PH detection models are unsuitable for detecting EH.’

Line 145: in the sentence, the phrase ' for field maize ' should be removed.

Line 186: the title of in the manuscript seems that there exists a writing error that is not relevant to the manuscript.

Line 216: Is α(θ) = 1.1 obtained from published literature? I suggest adding a reference to support the data source.

Line 222: 'And, to' is proposed to be changed with 'To'

Lines 222-223: is this step operated to remove the noise point cloud from the top layer? If so, and the top layer is not emphasized in the text, please state the operation clearly and simply

Line 256: '1-2' should be changed to 'one or two'

Lines 296-297: middle space deleted

Line 381: Capitalize the first letter of 'we'.

Line 449: The sentence would be more appropriate if changed to ' Various platforms and sensors acquire data in different ways.'

 Generally, this work was written in correct English and organized in a good shape. The conclusions and findings have merits in advancing related remote sensing sciences applied to agricultural section. I recommend a minor revision before accepting for publication.

Reviewer 3 Report

comments in attachment
